# Cognitive Behavioral Therapy plus Coping Management for Depression and Anxiety on Improving Sleep Quality and Health for Patients with Breast Cancer

**DOI:** 10.3390/brainsci11121614

**Published:** 2021-12-08

**Authors:** Hui-Ling Lai, Chun-I Chen, Chu-Yun Lu, Chiung-Yu Huang

**Affiliations:** 1Nursing Department, Tzu Chi University, Hualien 97004, Taiwan; snowjade@mail.tcu.edu.tw; 2Management College, I-Shou University, Kaohsiung 84001, Taiwan; eddychen@isu.edu.tw; 3Nursing Department, I-Shou University, Kaohsiung 82445, Taiwan; chuyun@isu.edu.tw

**Keywords:** sleep quality, cognitive behavioral therapy plus coping management, anxiety, depressive symptoms, quality of life

## Abstract

Cancer-related treatments may lead to side effects that undermine a patients’ quality of life (QOL). Although cognitive behavioral therapy plus coping management (CBTM) may appear to improve health-related QOL in cancer patients, limited documentation exists on the effectiveness of psychosocial interventions for patients with breast cancer (BC) during recovery. The purpose of this study was to examine the effectiveness of CBTM for sleep quality, anxiety, depression, and health among patients with BC. An experimental study was conducted to assess the efficacy of a CBTM intervention (experimental group = 36, control group = 34). The experimental group received a 12-week CBTM intervention focused on their identity, challenges, the replacement of dysfunctional beliefs, coping skills, relaxation, and rehabilitation exercises, while the control group received usual care. The follow-up evaluations were performed immediately after the intervention (T1), and at one (T2) and three months (T3). The generalized estimating equation (GEE) model showed significant effects from the CBTM intervention over time. The experimental group showed significant improvement in sleep quality, anxiety and depressive symptoms, and significant increases in their mental and physical QOL from baseline, T1, T2, and T3—except for the mental and physical QOL showing no significant change at T3—while the control group receiving usual care showed no changes over time. The results suggest that CBTM increases sleep quality, reduces anxiety and depressive symptoms, and enhances health-related QOL for participants. CBTM is efficacious and can be provided by nurses to enhance patients’ coping skills and consequently improve their QOL.

## 1. Introduction

Breast cancer (BC) is the cancer with the highest incidence in Taiwan [1], and women diagnosed with BC may be presented with a series of challenges and invasive treatments, often resulting in physical and emotional upset [2,3], and struggles with social isolation and adaptation [4]. Additionally, insomnia occurs in patients with BC quite often. The prevalence of insomnia in cancer populations ranges from 20% to 70% [5] and is usually higher than that found in non-cancer patients. Substantial comorbid emotional problems suggest that depression and anxiety are common, but the reported percentages of BC patients who experience depression and anxiety vary (36–55%) [4,6,7]. Different methodologies or treatments may contribute to various findings, and significant distress undoubtedly accompanies a diagnosis of BC [8].

Previous studies on the effectiveness of cognitive behavioral therapy (CBT) intervention for patients with BC have shown that CBT has benefits for psychological health in patients with BC [5] and in patients with depression [9]. CBT can enhance an individual’s ability to deal with their psychosocial and emotional problems, disease comorbidities, and promote their adaptability to their situations. Matthews, Grunfeld, and Turner conducted a meta-analysis of patients with BC who underwent surgery and highlighted that CBT was the most effective psychosocial intervention, improving anxiety and depression and increasing quality of life (QOL) among patients treated by the professionals in different medical fields.

### 1.1. Behavioral Therapy and Sleep Quality

Poor sleep quality commonly occurs in patients with BC and BC survivors during and after invasive treatments, including radiation or chemotherapy [7]. Insomnia refers to an incapability to fall asleep or waking up too early in the morning or night, which leads to lower daytime functioning; insomnia has been defined as a clinical syndrome involving the impairment of sleep quality that continues longer than one month and may consequently impact individual health [5]. CBT has been globally applied for insomnia to improve sleep quality and involves instructions for stimulus control, sleep restriction, and sleep hygiene that were developed by Bootzin [10]. Ballesio, Aquino, Feige et al. [11] reported significant effects of applying face-to-face CBT for insomnia on depressive symptoms in individuals with sleep disorders. There is a limited longitudinal study of the effect of CBT for insomnia in improving psychosocial outcomes in BC patients, and more research may be needed to confirm the effectiveness of CBT for insomnia that is comorbid with depression and anxiety among both early diagnosed BC patients and survivors of BC.

### 1.2. Rationale for the Use of CBTM Interventions

CBT for the treatment of sleep disturbances has been considered effective [12], but based on the previous studies with BC patients with insomnia who are comorbid with psychological problems, such as depressive or anxiety symptoms [5,13] and patients with depression (Williams et al., 2013), the potential effectiveness of CBT with such patients in Taiwan is unclear. In particular, BC patients usually require time to adapt to the threat of the disease and changes to their daily life, and we are unaware of anu evidence supporting the effectiveness of CBTM when used as part of the treatment of patients with BC in southern Taiwan. The objective of the present intervention was to determine whether CBTM is more effective than usual care in improving sleep quality in individuals with BC. The secondary objective was to assess whether CBTM is more effective than usual care in improving anxiety and depressive symptoms, mental QOL, and physical QOL. We hypothesized that a CBTM intervention (12 sessions over 3 months) would be effective in improving sleep quality and secondary outcomes (anxiety and depressive symptoms and health-related QOL [HRQOL]) from the baseline to the follow-up, including immediately after the intervention, and one and three months after the intervention.

## 2. Materials and Methods

### 2.1. Design

We followed the Consolidated Standards of Reporting Trials (CONSORT) guidelines [14]. A randomized controlled experimental study design was used to investigate the efficacy of CBTM for BC patients who met the inclusion criteria. Participants were randomly allocated to the intervention or control group by the investigators with the 12-session CBTM intervention applied by a licensed psychotherapist and an experienced registered nurse. Sleep quality, anxiety and depressive symptoms, and PQOL and MQOL were measured at baseline, post-intervention (T1), and at the 1-month (T2) and 3-month (T3) follow-ups.

### 2.2. Participants

Participants with BC who were receiving invasive therapy, i.e., either surgery or adjuvant therapy, were recruited between February 2016 and August 2017. The researcher received approval from the institutional review boards of the hospitals. The eligibility criteria included: (1) diagnosed with BC at 20 years of age or above, and (2) had an anxiety or depression HADS score of 7 or above and a sleep quality PSQI score higher than 5. The exclusion criteria included established neurological illness such as dementia or medical illness or severe physical impairments resulting in cognitive dysfunction. All participants provided written informed consent. To yield significant results with 80% power, we followed the recommendation of a medium effect size with an *f*^2^ equal to 0.5 by Cohen [15] and performed G-power 3.1 calculations; the adequate sample size was 35 for each group.

### 2.3. Data Collection

The sociodemographic information questionnaire was used to measure an individual’s characteristics. Data was collected on their age; education, which was reported as illiterate, elementary, junior high, senior high and college; income reported in New Taiwan currency (NT dollars); adjuvant therapy; breast cancer stage classified into four stages; and disease characteristics measured as duration in months.

#### 2.3.1. Variables and Measures

We used a set of questionnaires for face-to-face interviews, including a demographic information sheet and several measures as described below.

#### 2.3.2. Pittsburgh Sleep Quality Index

The Pittsburgh Sleep Quality Index (PSQI) [16] was applied to measure sleep quality; the PSQI consists of 19 items in seven sections and evaluates a participant’s perceived sleep efficiency, sleep duration, sleep latency, sleep quality, daytime dysfunction, and use of sleep pills. The global PSQI score is calculated by summing up the scores and ranges from 0 to 21. Participants with PSQI scores greater than 5 were identified as having poor sleep quality [16]. The Cronbach’s α that was used was 0.78 in Pan et al. [17] and 0.82 in the current study.

#### 2.3.3. Hospital Anxiety and Depression Scale

Anxiety and depressive symptoms were evaluated with the 14-item Hospital Anxiety and Depression Scale (HADS) [18]. The HADS consists of a 14-item self-report questionnaire comprising seven items for anxiety (HADS-A subscale) and seven items for depression (HADS-D subscale). Each item is scored from 0 to 3, and a total score of 8 or greater on each subscale indicates the incidence of depressive or anxiety symptoms. In this study, the Cronbach’s α coefficients were 0.79 and 0.82 for the HADS-A and HADS-D, respectively.

#### 2.3.4. Health Survey

The 36-item Short Form Survey (SF-36) was designed to be a universal indicator of health [19]; it consists of eight subscales that are appropriate for assessing the general health of individuals, including their physical function, role physical, bodily pain, general health, vitality, social functioning, role emotional, and mental health. HRQOL is usually separated into two categories, namely, physical QOL (PQOL) and mental QOL (MQOL), both of which were evaluated in this study. The higher the score on the scale, the better health. The Cronbach’s α coefficients in our study ranged from 0.74 to 0.83 for the SF-36 subscales.

### 2.4. Intervention

Participants were offered 12 sessions over 3 months. The control group received only the usual care from the nurses’ health education. The usual care was the routine care provided by the hospitals for breast cancer patients. The intervention was developed based on Lazarus and Folkman’s theory of stress and coping concepts and was culturally adapted by a counselor and researcher. The treatment agenda was applied to improve the coping skills and positive thinking in individuals with BC. The two main activities in the current study included motivational interviewing and active-coping treatments, which were administered in two groups of 17–18 patients during 12 weekly sessions, with each session lasting 2 h. Each session was structured as follows: agenda setting, homework planning, interactive presentations, group discussion, and feedback. The CBTM protocol included the following sessions applied by a licensed psychotherapist and an experienced registered nurse.

#### 2.4.1. Sessions One to Three: Introduction

Session One: In the introduction, the participants described their experiences of suffering BC and their physical symptoms, such as hot flashes and night sweats, and outlined their goals for treatment. The researchers encouraged the participants to participate in the group discussion. The group therapy included the encouragement of individual self-expression and the expression of empathy by the psychotherapist. Session Two: A cognitive behavior model was introduced, and the prevention of complications, lifestyle behavior changes, and coping stress management for patients with BC was discussed. Additionally, during the sessions, the psychotherapist introduced the concept of stress and coping strategies to help patients address their disease suffering, treatment journey, and management of their symptoms in different conditions. Session Three: The participants were encouraged to have self-awareness and share their positive and negative thinking. The psychotherapist and the researcher encouraged participants to exchange strategies to recognize their emotional reactions and behavioral techniques.

#### 2.4.2. Sessions Four to Six: Fighting with Symptoms

Session Four: The topics of fighting with the disease and therapy were discussed; the participants were encouraged to share their experiences and explore their cognitive behavior. In addition, patterns and coping strategies for the individuals to deal with the disease threat and treatment were discussed. Session Five: The participants shared their experiences of sleep disturbance comorbidity and learned how to relax and improve their sleep quality. Session Six: The participants explored pain and other symptoms, such as lymphatic swelling. The researcher instructed the participants on how to increase the effectiveness of their pain management.

#### 2.4.3. Sessions Seven to Nine: Body Image

Session Seven: The topics in this session included the practical strategies for positive thinking awareness and mindfulness stress reduction, as well as decision-making for self-healing, which promoted increased self-care of the body, mind, and spirit. Session Eight: The individual body image changes and concerns after suffering from the disease were discussed. The researcher focused on the reconstruction of the body. Session Nine: The viewpoints of the individuals and families about BC and the influence of the disease on family and interpersonal interactions were explored. The participants were exposed to the idea that understanding the behavior and thoughts of their family and friends would increase their awareness of fighting with the disease.

#### 2.4.4. Sessions Ten to Twelve: Resilience and Recovery

Session Ten: This session focused on increasing individual resilience. The psychotherapist encouraged the BC survivors to enhance their resilience. Session Eleven: The theme of this session was the spirit of a hero; the session focused on the growth after trauma, and the psychotherapist explored the influences of the disease on the different levels of the body, mind, and spirit. The participants were encouraged to cherish the opportunity to learn from their disease-suffering experience, including choosing to give a different value and meaning to the disease after experiencing trauma. Session Twelve: This final session of the CBTM intervention focused on the Lifestyles of Health and Sustainability (LOHAS); the participants were encouraged to retrospectively reflect and look ahead to the future to explore their increased capabilities for self-care.

In summary, all sections covered cognitive restructuring, the replacement of dysfunctional beliefs, appraisal skills, and training in problem-solving techniques. The psychotherapist and the nursing professional worked with several scenarios that commonly occur in patients with BC. Additionally, the participants shared how they usually coped with difficult situations such as sleep disturbances, anxiety, or hot flashes. The nurse suggested that the participants increase their motivation for self-achievement and attend activities in their social networks. During the therapy process, the psychotherapist listened with empathy, presented personal feedback, helped the participants improve their assertiveness and communication skills, and suggested that the participants access social support.

### 2.5. Ethical Considerations

The study protocol was approved by the Institutional Ethics Review Board of the hospital in southern Taiwan (EMRP03104N). All participants provided written informed consent before the study was conducted.

### 2.6. Statistical Analysis

A descriptive analysis was implemented; the means, standard deviations (SDs), and frequency distributions were calculated for the quantitative variables. Repeated measures analyses of the variance were conducted to test the effects of CBTM on depression, anxiety, PSQI global scores, and QOL. The intervention was provided for 12 weeks, and the data were collected at baseline, immediately after the 12 weeks of treatment, and one and three months after the intervention. Participants were free to drop out during the intervention. The statistical analysis was performed using the SPSS Predictive Analytics Software package (PASW Statistics 23). Pre-intervention comparisons of the sample characteristics were conducted using an analysis of variance and χ^2^ tests. All exams were two-tailed, and any factor differing between the groups was considered a covariate.

The generalized estimating equations (GEEs) were applied to further evaluate the effectiveness of CBTM for improving anxiety, depression, and QOL before and after the 12-week CBTM, and one and three months later. GEEs are widely used to analyze the interconnected longitudinal data and can also be used to analyze the clinical data with a non-normal distribution or missing data among groups [20,21]. We selected a first-order autoregressive AR(1) as the working correlation structure in the GEE for the intracluster correlation structure data [22]. Moreover, we applied the GEEs through a repeated measure analysis of changes in sleep quality, anxiety, depression, and HRQOL at baseline, T1, and T2 to identify the between-group differences over time (T3).

## 3. Results

The study flow diagram is shown in Figure 1. Of the 207 BC patients screened from the initial study, 130 were eligible, 77 did not meet the inclusion criteria, 55 did not want to participate in the CBTM for further study, and 36 were allocated to the CBTM group and completed the 12-week therapy until August 2017. Thirty-four patients in the control group received only usual care, and information was collected from these patients during their regular general surgical department follow-ups.

### 3.1. Participant Characteristics

Table 1 shows that the demographic characteristics between the experimental and control groups were equivalent regarding all personal characteristics at baseline. The mean age of the 70 participants was 56.43 (SD = ±10.42) years old. Most participants had either high school qualifications or above; 23 of the subjects had completed high school, and approximately 14 were college graduates. Seventy adults had a time since BC diagnosis of at least 12 months, and most of the participants’ household incomes were between 25,000 and 50,000 NT dollars per month. The average duration of BC was approximately 16 months (SD = ±18.26) and did not differ between the two groups, and 46 of the participants had stage I or II BC. The majority of the participants (59 BC patients) had received adjuvant treatments.

### 3.2. Differences in the Outcome Variables between the Two Groups at Baseline, Post-Intervention (T1), 1-Month (T2) and 3-Month (T3) Follow-Up

At baseline, sleep quality was highly prevalent, with a mean PSQI score of 7.07 (SD = 2.47); anxiety and depressive symptoms were also prevalent, with a mean HADS-A score of 9.19 (SD = 2.7) and a mean HADS-D score of 15.17 (SD = 1.56). Thirty-six women received the CBTM intervention for sleep quality, anxiety, and depressive symptoms, and 34 women had usual care. There was no significant difference between the CBTM and control group in the sleep quality of PSQI scores (7.50 vs. 6.62, *p* = 0.14, 95% CI: −0.29 to 2.05). After the 12-week CBTM intervention (T1), the PSQI score was decreased to 5.33 (SD 1.84) in the CBTM group compared with 6.18 (SD 3.23) in the control group (*p* = 0.19, 95% CI: −2.08 to 0.40). However, one month after CBTM (T2), there was a significant difference of the PSQI scores between the two groups (*p* < 0.001, 95% CI: −3.6 to −1.24) (Table 2). Three months after the CBTM (T3), the impairment of sleep quality score had decreased to 3.56 (SD 1.23) in the CBTM group and was 6.82 (SD 3.23) in the usual care group (Table 2). As shown in Figure 2, there was a steady reduction of the PSQI from 7.5 (at baseline) to 3.56 (T3) in the CBTM group (3.94 points decreased), and slight increase from 6.62 (at baseline) to 6.82 (T3) in the control group (0.2 point increased).

Regarding anxiety (as shown in Figure 3), there was no significant difference between the CBTM and control groups at baseline (9.18 vs. 9.26, *p* = 0.92, 95% CI: −1.36 to 1.23) and after the CBTM intervention (T1) (7.64 vs. 7.53, *p* = 0.86, 95% CI: −1.08 to 1.30). However, there was a significant difference between groups in the anxiety primary outcome variable at T2 (*p* = 0.02, 95% CI: −2.86 to −0.24) and T3 (*p* < 0.0001, 95% CI: −3.86 to −1.42), with a greater reduction in the anxiety score from baseline in the CBTM group than in the usual care group. In T3, the anxiety score showed a significant reduction to 5.92 (SD 1.63) in the CBTM group compared with those in the usual care group, which was 8.56 (SD 3.27) (Table 2). As shown in Figure 3, there was a steady reduction in anxiety from 9.18 (at baseline) to 5.92 (T3) in the CBTM group (3.26 points decreased), and 9.26 (at baseline) to 8.56 (T3) in the control group (0.7 point decreased).

Regarding the depressive symptoms of the BC patients (as shown in Figure 4), there was no significant difference between the CBTM and control groups at baseline (15.31 vs. 15.03, *p* = 0.47, 95% CI: −0.48 to 1.03) and T1 (12.50 vs. 13.15, *p* = 0.09, 95% CI: −1.39 to 0.10). However, there was a significant difference between the groups in the depression score at T2 (*p* < 0.0001, 95% CI: −2.53 to −1.08) and T3 (*p* < 0.0001, 95% CI: −5.92 to −3.99) (Table 2). The depression score showed a significant reduction to 9.72 (SD 1.70) in the CBTM group compared with the usual care group’s 14.68 (SD 2.32) in T3 (Table 2). As shown in Figure 4, there was a steady reduction in depression from 15.31 (at baseline) to 9.72 (T3) in the CBTM group (5.59 points decreased).

Regarding health-related QOL, mental QOL (MQOL) and physical QOL (PQOL) were assessed. For MQOL, there was no significant difference between the CBTM and control groups at baseline (56.39 vs. 56.06, *p* = 0.89, 95% CI: −4.56 to 5.22) and T1 (57.50 vs. 55.18, *p* = 0.29, 95% CI: −2.03 to 6.68). However, there was a significant difference between groups in the MQOL score at T2 (*p* < 0.0001, 95% CI: 2.06 to 9.88) and T3 (*p* < 0.0001, 95% CI: 5.52 to 14.0) (Table 2). As shown in Figure 5, there was a steady increase in MQOL from 56.39 (at baseline) to 63.22 (T3) in the CBTM group (6.83 points increased), and a reduction from 56.06 (at baseline) to 53.41 (T3) in the control group (3.09 point decreased). For PQOL, there was no significant difference between the two groups from baseline, T1, T2, and T3 (Table 2). As shown in Figure 6, there was a steady increase in PQOL from 55.75 (at baseline) to 59.28 (T3) in the CBTM group (3.53 points increased), and a reduction from 58.76 (at baseline) to 55.32 (T3) in the control group (3.44 point decreased).

### 3.3. Generalized Estimating Equations

The test of the GEE model showed significant effects from CBTM in the impairment of sleeping quality (PSQI), anxiety (HAD-A), depression (HAD-D), MQOL, and PQOL from baseline through T3 (Table 3). For PSQI, the CBTM group represented a steady reduction of the slope as *B* = −4.15 (standard error, SE 0.38, *p* < 0.001) from baseline to T1, *B* = −2.43 (SE 0.38, *p* < 0.001) from T1 to T2, and *B* = −0.87 (SE 0.38, *p* = 0.02) from T2 to T3. For example, the CBTM group for the PSQI estimated score at T3 was 3.56 (intercept) + 3.27 (CBTM baseline) + (−4.15) (CBTM group × baseline) + (−2.43) (CBTM group × T1) + (−0.87) (CBTM × T2) = −0.62. The estimated score of the PSQI at T3 for the control group, on the other hand, was 3.56 + 3.94 (control group baseline) + 1.78 (T1) + 0.78 (T2) = 10.06. For anxiety, the CBTM group reflected a steady reduction of the slope as *B* = −2.58 (SE 0.49, *p* < 0.001) from baseline to T1, *B* = −2.75 (SE 0.49, *p* < 0.001) from T1 to T2, and *B* = −1.09 (SE 0.49, *p* = 0.03) from T2 to T3. For depression, the CBTM group reported a steady reduction of the slope as *B* = −5.23 (SE 0.52, *p* < 0.001) from baseline to T1, *B* = −4.31 (SE 0.52, *p* < 0.001) from T1 to T2, and *B* = −3.15 (SE 0.52, *p* = 0.03) from T2 to T3.

For MQOL, the CBTM group reported a steady improvement of the slope as *B* = 9.48 (SE 2.15, *p* < 0.001) from baseline to T1, and *B* = 7.49 (SE 2.15, *p* < 0.001) from T1 to T2. However, the long-term effect of CBTM on MQOL was *B* = 3.84 (SE 2.15, *p* = 0.07) from T2 to T3, which was insignificant. For PQOL, the CBTM group revealed a steady improvement of the slope as *B* = 6.97 (SE 1.14, *p* < 0.001) from baseline to T1, *B* = 2.99 (SE 1.14, *p* = 0.01) from T1 to T2. However, the long-term effect of CBTM on PQOL was *B* = 1.64 (SE 1.14, *p* = 0.15) from T2 to T3, which was insignificant.

Furthermore, most of the changes in the corresponding B values of the interaction effects from baseline to T1, T2, and T3 indicated the effects of the CBTM occurred over time. These findings indicate that the strong effects were influenced from baseline to T1, T1 to T2, and less from T2 to T3, particularly in PQOL and MQOL.

## 4. Discussion

BC is one of the leading causes of global morbidity and mortality in Taiwan. Fortunately, with early detection and the expansion of treatment, a greater number of BC patients can survive longer with cancer. The increased survival rates for BC indicate that more people will accept repeated invasive therapy, despite the side effects. Sleep disturbances are one of the most frequent complaints that cancer survivors suffer, which would affect their mental and physical well-being [12,23,24].

This study aimed to examine the effectiveness of a CBTM intervention to treat sleep disturbances in women with BC. Not surprisingly, after three months of the intervention, there were significantly greater improvements in the several outcome variables among individuals who received the intervention than among the control group participants who received their usual care. Additionally, a significantly larger proportion of the CBTM patients than control patients experienced an increase in sleep quality between posttreatment (T1) and follow-up (T2 & T3), indicating that the intervention benefits persisted up to 3 months after the CBTM. Notably, the effect of CBTM had slightly decreased at T3; this finding suggests that CBTM should be continued for a longer duration or a follow-up to remind patients about the implementation of such an approach. These significant results are consistent with the previous studies [24]. In a meta-analysis study by Okajima and Inoue [25], the effect of CBT for insomnia was supported in people with psychiatric disorders and medical diseases, which suggested the causal effect of CBT for sleep quality. According to the present results indicating significant improvement in the outcome variables, a combined intervention on sleep hygiene and relaxation skills led by nurses could be implemented in cancer clinics for BC patients.

The results also showed a greater decrease in anxiety and depressive symptoms among the patients who received the CBTM intervention than among the patients who received usual care. The finding showed that CBTM had significant effects in decreasing anxiety over time, but the strength of the effect at T3 was lower than the effect of usual care at T3 for the control participants. These results are similar to those of previous studies [26,27]. According to a study by Peoples et al. [28] CBT for insomnia could manage sleep disturbances, increase sleep quality, and further reduce depressive symptoms in cancer survivors. Furthermore, our study findings are consistent with Ho et al. [24] and Xiao et al.’s studies [29], which show that there were strongly significant decreases in the depressive symptoms over time in the CBTM group. In summary, literature results showed that CBT is effective in managing sleep quality and improving mental well-being (i.e., depression or anxiety). Sleep quality and mental well-being are often correlated with each other; poor mental well-being is usually associated with poor sleep quality, and vice versa. Further research is needed to examine the association among CBT, sleep quality, depression, and anxiety.

Our intervention provided 12 weeks of CBTM, combining stress management, coping skills, and relaxation protocol, which was longer than Groarke et al.’s brief CBT intervention [27]; thus, the effect of the CBTM lasted three months after the intervention. For participants in the control group, their anxiety and depression remained prevalent throughout the study period. This is correspondent with previous studies; BC patients often continue to report high prevalence of anxiety and depression throughout the treatment periods [30].

Regarding QOL, we found that the MQOL was significantly improved in the experimental group than in the control group, but there was a positive time effect at T1 and T2; the PQOL was significantly greater in the experimental group than in the control group in the GEE results of Table 3, but the time effect was not significant at T3. For the PQOL outcome in Table 3, the control group revealed a significantly higher score (*B* = −3.53, *p* < 0.001) compared with the CBTM group (*B* = −3.95, *p* = 0.19) at baseline. The slop of the PQOL mean score gradually declined from baseline to T1 (*B* = −2.53, *p* = 0.002) and T2 (*B* = −0.58, *p* = 0.47). Compared with the CBTM group, the intervention effect was *B* = 6.97 at T1 (CBTM group*baseline, *p* < 0.001) and *B* = 2.99 at T2 (CBTM*T2, *p* = 0.009). This corresponds with the data in Figure 5 that PQOL declines over time in the control group, while it improves continuously in the CBTM group. Simply put, the estimated PSQI at T3 for the CBTM group was 59.28 + (−3.95) (CBTM baseline) + 6.97 (CBTM group*baseline) + 2.99 (CBTM group*T1) + 1.64 (CBTM group*T2) = 66.93. The estimated PSQI at T3 for the control group was 59.28 + (−3.53) (control baseline) + (−2.53) (T1) + (−0.58) (T2) = 52.64. A longer follow-up period may be necessary to examine the effectiveness of the intervention. Furthermore, for participants in the control group, their scores in MQOL and PQOL slightly declined from baseline through T3, suggesting that usual care might be inadequate in adjusting or managing the change or stress coming with BC.

One limitation of this study was the self-report procedures. Physiological and self-report measures were used at baseline and 12 weeks after the intervention. Additionally, we did not control for all possible confounding factors related to medication treatment, such as the use of herbal drugs to manage sleep problems and the use of adjuvant hormone therapy, which may have had related effects. Furthermore, the choice of a control group with standard care is suboptimal and it does not exclude the presence of placebo effects. Sleep hygiene or other health behaviors may also influence outcomes, but these variables were not controlled in the analyses. Our findings suggest that CBTM treatment can be provided by nursing professionals during patients’ outpatient visits, and that the intervention could be provided to both BC patients and survivors to improve their QOL.

In sum, these results highlight the value of CBTM for treating poor sleep quality during cancer care to reduce comorbid anxiety and depression. Additionally, we may also focus on CBT for poor sleep quality in future research, as evidence on the use of CBT for sleep disorders has been provided—but not evidence of the use of CBT with breast cancer survivors.

## 5. Conclusions

Our findings suggest that CBTM is safe and effective in helping women manage the stress and negative symptoms after BC treatment, and that it has additional benefits for sleep, mental health, and QOL. Despite its limitations, this study increases the understanding of the effects of psychoeducation therapy on sleep quality and mental health, including anxiety and depression. This CBTM intervention may also provide support and promote adaptation skills among BC patients. Future studies may focus on CBT combined with sleep hygiene and quality training. Further programmatic research in the form of rigorously designed randomized trials will help determine the efficacy of CBTM in increasing sleep quality in BC patients, and patients treated by professionals in other medical fields or even the application of internet CBTM for convenient communication during the coronavirus pandemic. Our study showed that acute poor sleep quality can be successfully treated using CBTM, and we can further develop a protocol focusing on addressing the sleep problems in BC patients.

## Figures and Tables

**Figure 1 brainsci-11-01614-f001:**
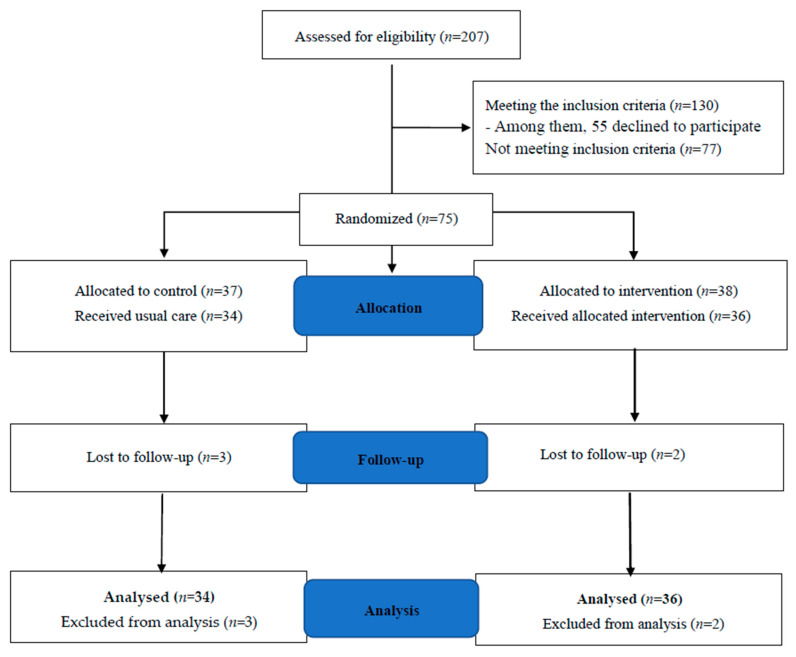
Flow Diagram of CBTM in breast cancer (BC) patients. Note. Intervention was initiated at baseline in the intervention group and coping management. Control group patients received usual care from medical and nursing staff only throughout the study period. Outcomes, i.e., sleep quality, anxiety, depression, physical quality of life (PQOL), and mental quality of life (MQOL), were measured at before, after, and one and three months after cognitive behavioral therapy plus coping management (CBTM).

**Figure 2 brainsci-11-01614-f002:**
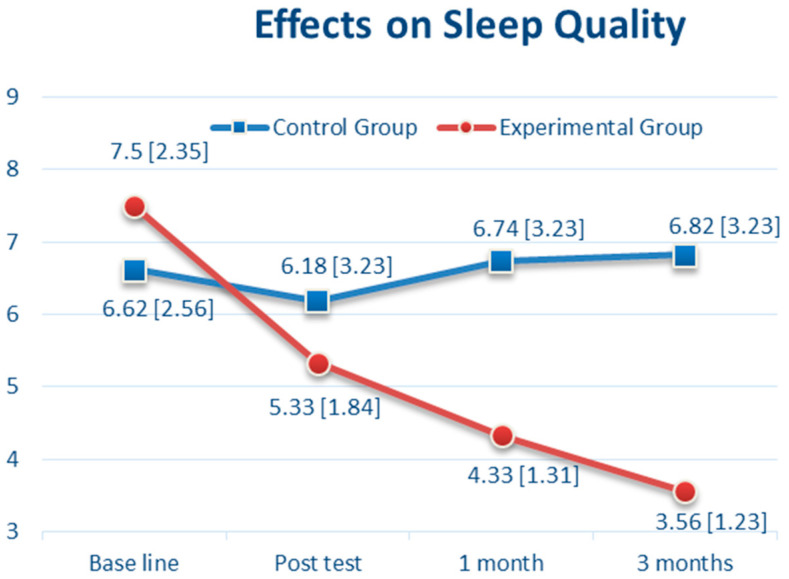
The Effects over Time of Cognitive Behavioral Therapy (CBT) on Sleep Quality.

**Figure 3 brainsci-11-01614-f003:**
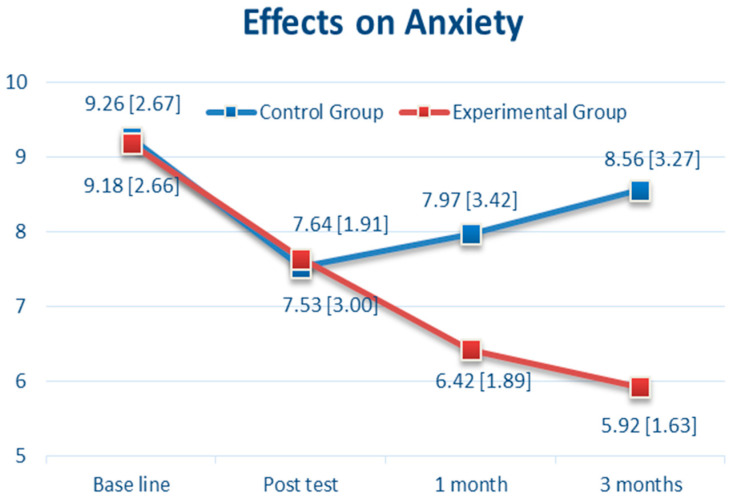
The Effects over Time of Cognitive Behavioral Therapy (CBT) on Anxiety.

**Figure 4 brainsci-11-01614-f004:**
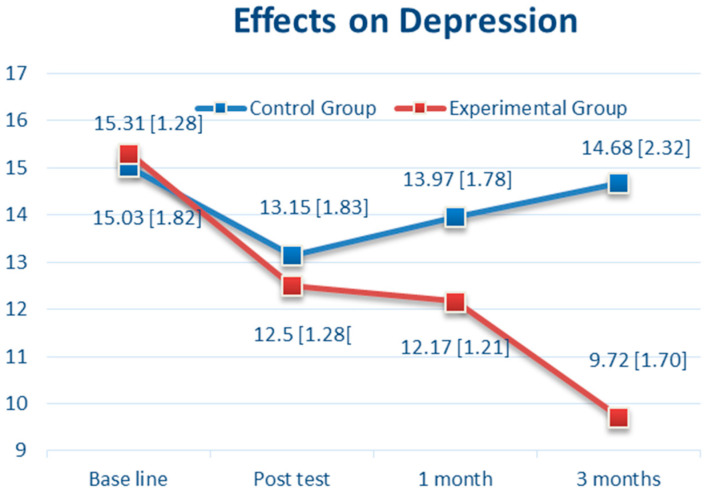
The Effects over Time of Cognitive Behavioral Therapy (CBT) on Depression.

**Figure 5 brainsci-11-01614-f005:**
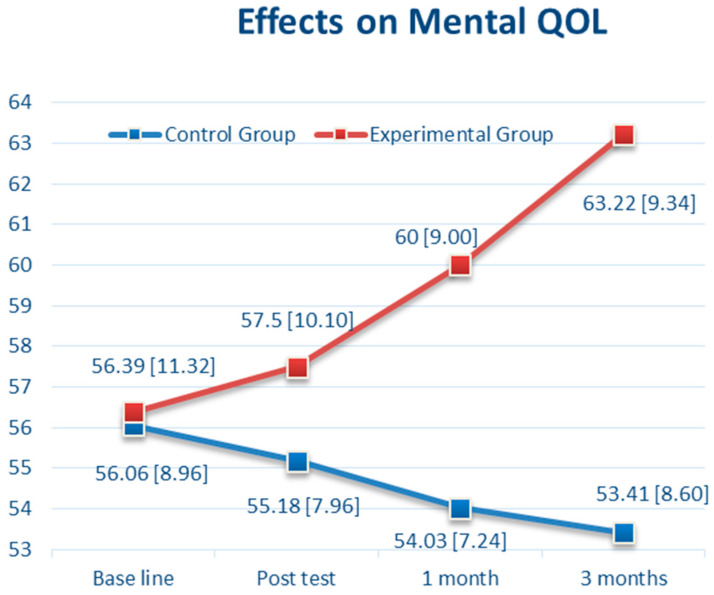
The Effects over Time of Cognitive Behavioral Therapy (CBT) on Mental Quality of Life (MQOL).

**Figure 6 brainsci-11-01614-f006:**
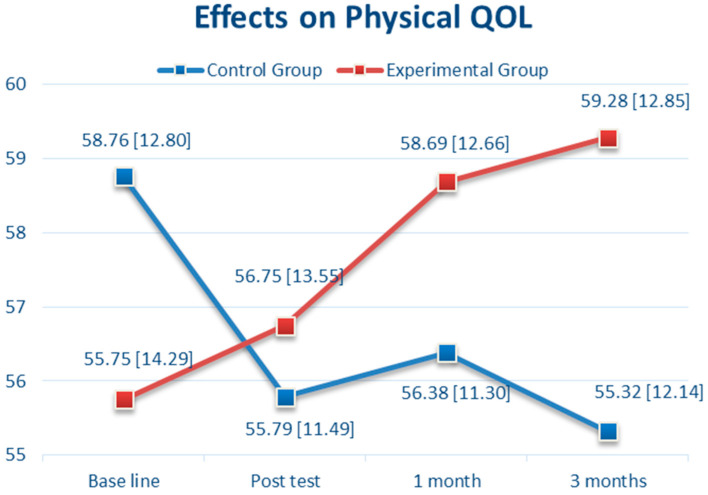
The Effects over Time of Cognitive Behavioral Therapy (CBT) on Physical Quality of Life (PQOL).

**Table 1 brainsci-11-01614-t001:** The Comparison of Demographic and Clinical Features between Two Groups.

	Control Group	CBT Group		
	*n* = 34	*n* = 36	Total	*p*-Value
Age	53.24 ± 9.67	50.28 ± 9.20	56.43 ± 10.42	0.19
Education				
Illiterate	3	3	6	0.21
Elementary	10 4	5	15	
Junior high	4	8	12	
Senior high	13	10	23	
College	4	10	14	
Income				0.30
(Monthly NT)				
<25,000	13	5	18	
25,001~50,000	13	10	23	
50,001~75,000	2	14	16	
75,001~100,000	4	4	8	
>100,000	2	3	5	
Adjuvant therapy				0.92
without	6	5	11	
with	28	31	59	
Stage of breast cancer				0.75
I	10 13	8	18	
II	13	15	28	
III	8	8	16	
IV	4	4	8	
Duration (month)	19.85 ± 23.17	12.24 ± 11.04	15.94 ± 18.26	0.08

**Table 2 brainsci-11-01614-t002:** Mean (SD) Differences from Baseline to Post CBTM, 1-Month & 3-Months Follow-up.

	Control (*n* = 34)	CBTM (*n* = 36)	Total	*p*-Value	C.I. of 95%
	(Mean [SD])	(Mean [SD])	*n* = 70		
Sleep Quality					
Baseline	6.62 (2.56)	7.50 (2.35)	7.07 (2.47)	0.14	[−0.29 to 2.05]
Post-CBTM (T1)	6.18 (3.23)	5.33 (1.84)	5.74 (2.62)	0.19	[−2.08 to 0.40]
1 Month (T2)	6.74 (3.23)	4.33 (1.31)	5.50 (2.71)	0.00	[−3.60 to −1.24]
3 Month (T3)	6.82 (3.23)	3.56 (1.23)	5.14 (2.91)	0.00	[−4.46 to −2.08]
Anxiety					
Baseline	9.26 (2.67)	9.18 (2.66)	9.19 (2.70)	0.92	[−1.36 to 1.23]
Post-CBTM (T1)	7.53 (3.00)	7.64 (1.91)	7.59 (2.48)	0.86	[−1.08 to 1.30]
1 Month (T2)	7.97 (3.42)	6.42 (1.89)	7.17 (2.83)	0.02	[−2.86 to −0.24]
3 Month (T3)	8.56 (3.27)	5.92 (1.63)	7.20 (2.87)	0.00	[−3.86 to −1.42]
Depression					
Baseline	15.03 (1.82)	15.31 (1.28)	15.17 (1.56)	0.47	[−.48 to 1.03]
Post-CBTM (T1)	13.15 (1.83)	12.50 (1.28)	12.81 (1.59)	0.09	[−1.39 to 0.10]
1 Month (T2)	13.97 (1.78)	12.17 (1.21)	13.04 (1.76)	0.00	[−2.53 to −1.08]
3 Month (T3)	14.68 (2.32)	9.72 (1.70)	12.13 (3.20)	0.00	[−5.92 to −3.99]
Mental QOL					
Baseline	56.06 (8.96)	56.39 (11.32)	56.23 (0.17)	0.89	[−4.56 to 5.22]
Post-CBTM (T1)	55.18 (7.96)	57.50 (10.10)	56.37 (9.14)	0.29	[−2.03 to 6.68]
1 Month (T2)	54.03 (7.24)	60.00 (9.00)	57.10 (8.67)	0.00	[2.06 to 9.88]
3 Month (T3)	53.41 (8.60)	63.22 (9.34)	58.46 (0.20)	0.00	[5.52 to 14.10]
Physical QOL					
Baseline	58.76 (12.80)	55.75 (14.29)	57.21 (3.58)	0.36	[−9.50 to 3.47]
Post-CBTM (T1)	55.79 (11.49)	56.75 (13.55)	56.29 (2.51)	0.75	[−5.05 to 6.96]
1 Month (T2)	56.38 (11.30)	58.69 (12.66)	57.57 (1.99)	0.42	[−3.42 to 8.05]
3 Month (T3)	55.32 (12.14)	59.28 (12.85)	57.36 (2.58)	0.19	[−2.02 to 9.92]

Abbreviations: SD: Standard deviation; CBTM: Cognitive behavioral therapy plus coping management; C.I.: Confidence interval; PQOL: Physical related quality of life; MQOL: Mental related quality of life. T1, T2 and T3 are follow-up evaluations were performed immediately after the intervention, one and three months, respectively.

**Table 3 brainsci-11-01614-t003:** Effects of the CBTM on Patient’s PSQI, Anxiety, Depression, and PQOL, & MQOL in the Before, Post-Intervention (T1), 1-Month (T2), and 3-Month (T3) Follow-up (*n* = 70).

PSQI	*B*	SE	95% Wald C.I.		Wald χ^2^	*p*
			Lower	Upper		
Intercept	3.56	0.41	2.75	4.37	73.93	<0.001
CBTM group (baseline)	3.27	5.93	2.11	4.43	30.34	<0.001
Control group baseline	3.94	0.26	3.43	4.46	227.15	<0.001
T_1_	1.78	0.26	1.27	2.29	46.14	<0.001
T_2_	0.78	0.26	0.27	1.29	8.83	0.003
CBTM group × baseline	−4.15	0.38	−4.89	−3.41	122.15	<0.001
CBTM group × T_1_	−2.43	0.38	−3.16	−1.69	41.70	<0.001
CBTM group × T_2_	−0.87	0.38	−1.60	−0.13	5.32	0.021
Anxiety						
Intercept	5.92	0.44	5.06	6.78	182.32	<0.001
CBTM group (baseline)	2.64	0.63	1.41	3.87	17.66	<0.001
Control group baseline	3.19	0.34	2.53	3.86	87.54	<0.001
T_1_	1.72	0.34	1.05	2.39	25.45	<0.001
T_2_	0.50	0.34	−0.17	1.17	2.15	0.143
CBTM group × baseline	−2.58	0.49	−3.54	−1.62	27.67	<0.001
CBTM group × T_1_	−2.75	0.49	−3.71	−1.79	31.55	<0.001
CBTM group × T_2_	−1.09	0.49	−2.05	−0.13	4.94	0.026
Depression						
Intercept	9.72	0.28	9.17	10.27	1204.35	<0.001
CBTM group (baseline)	4.95	0.40	4.17	5.74	151.90	<0.001
Control group baseline	5.58	0.36	4.88	6.29	239.81	<0.001
T_1_	2.78	0.36	2.07	3.48	59.36	<0.001
T_2_	2.44	0.36	1.74	3.15	45.97	<0.001
CBTM group × baseline	−5.23	0.52	−6.24	−4.22	102.22	<0.001
CBTM group × T_1_	−4.31	0.52	−5.32	−3.29	69.32	<0.001
CBTM group × T_2_	−3.15	0.52	−4.16	−2.14	37.08	<0.001
MQOL						
Intercept	63.22	1.53	60.23	66.22	1712.59	<0.001
CBTM group (baseline)	−9.81	2.19	−14.11	−5.51	20.03	<0.001
Control group baseline	−6.83	1.50	−9.77	−3.90	20.80	<0.001
T_1_	−5.72	1.50	−8.66	−2.79	14.58	<0.001
T_2_	−3.22	1.50	−6.16	−0.29	4.62	0.032
CBTM group × baseline	9.48	2.15	5.27	13.69	19.44	<0.001
CBTM group × T_1_	7.49	2.15	3.27	11.70	12.13	<0.001
CBTM group × T_2_	3.84	2.15	−0.37	8.05	3.19	0.074
PQOL						
Intercept	59.28	2.12	55.13	63.42	785.61	<0.001
CBTM group (baseline)	−3.95	3.03	−9.90	1.99	1.70	0.193
Control group baseline	−3.53	0.80	−5.09	−1.96	19.55	<0.001
T_1_	−2.53	0.80	−4.09	−0.96	10.04	0.002
T_2_	−0.58	0.80	−2.15	0.98	0.54	0.465
CBTM group × baseline	6.97	1.14	4.73	9.21	37.06	<0.001
CBTM group × T_1_	2.99	1.14	0.76	5.24	6.86	0.009
CBTM group × T_2_	1.64	1.14	−0.60	3.89	2.06	0.151

Abbreviations: CBTM: Cognitive behavioral therapy plus coping management; C.I.: Confidence interval. B: Beta coefficient; SE: Standard error; PSQI: Pittsburgh Sleep Quality Index; PQOL: Physical Quality of Life; MQOL: Mental Quality of Life.

## Data Availability

Data sharing not applicable.

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
