# Peer review of "Cognitive Behavioral Therapy plus Coping Management for Depression and Anxiety on Improving Sleep Quality and Health for Patients with Breast Cancer"

_brainsci, 2021, doi:10.3390/brainsci11121614_

Round 1
Reviewer 1 Report
Thank you for the opportunity to read this interesting and important study. I think overall the language should be changed so it is clear whether these women are still undergoing treatments or are finished treatment and breast cancer survivors, and if so, how long post treatment? It was not necessarily clear to me in reading this what stage these women were at so this should be clarified throughout.
Introduction
I would say that generally the introduction could use more transitional statements as it reads somewhat choppy. One thing that I don't think seems clear is the role of sleep disturbance in breast cancer - is insomnia the result of the cancer treatment itself? Does cancer diagnosis and treatment lead to emotional disorders like mood and anxiety, which in turn, manifests as sleep disturbance? Sleep disturbance (e.g., insomnia) is a symptom of mood and anxiety concerns. I think this could be more clear in your introduction.
Methods
It was not clear to be what exactly the control group is - what is a waitlist control? Did they receive a different intervention? If so what was it? I saw in the results it is later described as treatment as usual, does that include any psychosocial supports?
Discussion
In reading the discussion I am still wondering whether sleep is independent from depression and anxiety, is it the improvement in those symptoms that in turn improves sleep? are they all independent?
Author Response
Table of Responses to Comments by Reviewer 1
|
Introduction: |
|
|
Suggestion 1 |
I would say that generally the introduction could use more transitional statements as it reads somewhat choppy. One thing that I don't think seems clear is the role of sleep disturbance in breast cancer - is insomnia the result of the cancer treatment itself? Does cancer diagnosis and treatment lead to emotional disorders like mood and anxiety, which in turn, manifests as sleep disturbance? Sleep disturbance (e.g., insomnia) is a symptom of mood and anxiety concerns. I think this could be more clear in your introduction. |
|
Response 1 |
Thanks for the suggestion, we have asked professional editing organization to rephrase our manuscript and add more description about the role of insomnia in the introduction. The certificate for editing is attached to this letter. |
|
Methods: |
|
|
Suggestion 2 |
It was not clear to be what exactly the control group is - what is a waitlist control? Did they receive a different intervention? If so what was it? I saw in the results it is later described as treatment as usual, does that include any psychosocial supports? |
|
Response 2 |
Thanks for the comments. The control group receive a usual care by nurses’ health education with non-specific psychosocial supports. In this research, the experiment design does not include the wait list control; therefore, there is no follow-up intervention procedure. |
|
Discussion: |
|
|
Suggestion 3 |
In reading the discussion I am still wondering whether sleep is independent from depression and anxiety, is it the improvement in those symptoms that in turn improves sleep? are they all independent? |
|
Response 3 |
Thanks for the comments. A meta-analysis study by Okajima and Inoue [25], the effect of CBT for insomnia was supported in people with psychiatric disorders and medical diseases, which suggested the causal effect of CBT (line 366-368). According to a study by Peoples et al., [28], CBT for insomnia could manage sleep disturbances and thus reduce depression in cancer survivors (line377-378). |

Reviewer 2 Report
This study assessed the effectiveness of a cognitive behavioral therapy (CBT) intervention for the treatment of the psychological side-effects of breast cancer. A twelve-week group therapy program reduced anxiety, depression and insomnia and improved quality of life in treated patients compared with a control group receiving usual care. Group differences in most of these measures even increased with time after the completion of treatment, suggesting sustained effects of CBT.
The study addressed a clinically relevant topic in a sufficiently large sample. The effects are clear and convincing, and, apart from some minor language problems, the manuscript is well written. My main reservation is the lack of novelty, given that the effectiveness of CBT for the treatment of insomnia, anxiety and depression is fairly well established.
Major comments
Final paragraph of Introduction: As there is little doubt about the effectiveness of CBT, the authors should highlight the specific contribution of this study. The fact that the effectiveness of CBT has not yet been demonstrated for breast cancer patients in southern Taiwan is not a convincing rationale for a study.
The control group received usual care. Therefore the specific contribution of CBT for symptom reduction remains unclear. Some kind of "placebo" treatment (e.g., support group meetings without the specific elements of CBT) would have helped to control the unspecific effects of being included in a treatment program, receiving attention and the possibility to talk about one's problems etc. This shortcoming should be addressed by the authors.
For sleep disturbances, anxiety and depression, treated patients did not differ from the controls at T1 (post-test), but only later, at 1 month and 3 months post-test. What could be the reason for this delayed response to treatment?
Was CBT administered by nurses without support by a qualified psychotherapist? Please clarify!
Minor comments
l. 98: depression/anxiety/sleep quality scores: which instruments were they based on?
l. 215: what does "AR(1)" stand for?
Results: for the analyses of the main outcome measures, testing for interactions between the factors "group" and "time point" would be appropriate and informative.
A measure of variance (standard deviation) should be added to the graphs in Figures 2-5.
Table 2 / Discussion (l. 373): According to the Table, there were no significant group differences at any time point for PQOL. However, the contrary is stated in the Discussion ("PQOL was significantly greater in the experimental group..."). Please clarify!
Language
l. 22: "over times" should be "over time"
l. 252 and several other places in the Results section: "as showed" should be "as shown", "a stead reduction" should be "a steady reduction"
l. 253 here "steady reduction" does not apply to the control group.
l. 264, l. 267 and several other places in the Results section: "in baseline", "in T2"etc should be "at baseline", "at T2" etc.
l. 316 and several other places in the Results section: "slop" should be "slope"
l. 343 "...physical health well-being" should be "...physical well-being"
l. 371: "strongly significantly better" sounds odd
l. 381: "manipulate" should probably be "control"
Author Response
Table of Responses to Comments by Reviewer 2
|
Introduction: |
|
|
Suggestion 1 |
The study addressed a clinically relevant topic in a sufficiently large sample. The effects are clear and convincing, and, apart from some minor language problems, the manuscript is well written. My main reservation is the lack of novelty, given that the effectiveness of CBT for the treatment of insomnia, anxiety and depression is fairly well established. |
|
Response 1 |
Thanks for the comments. We have asked professional language editing institute to improve the manuscript and the certificate is attached to this letter. The CBT of intervention may have contents fit to the southern participants of Taiwan, which is different from the previous literature. As our research team is in this area, we do believe we have both clinical and theoretical contribution to academic society. |
|
Major comments |
|
|
Suggestion 2 |
Final paragraph of Introduction: As there is little doubt about the effectiveness of CBT, the authors should highlight the specific contribution of this study. The fact that the effectiveness of CBT has not yet been demonstrated for breast cancer patients in southern Taiwan is not a convincing rationale for a study. |
|
Response 2 |
Thanks for the comments. Breast cancer (BC) is the cancer with the highest incidence in Taiwan. Healthcare services for cancer related treatments are expensive. The benefit package of cancer treatments (including surgery, chemotherapy, radiotherapy, and medications) is primarily covered by the National Health Insurance in Taiwan. In 2020, there were 148,734 BC patients who were covered by the National Health Insurance for their cancer treatments. Furthermore, the variety of health care services for cancer treatments is not equally distributed in Taiwan, and southern Taiwan. CBT is out-of-pocket health service, and it is less likely to be available in southern Taiwan. To examine the effect of CBTM for insomnia, QOL, and mental well-being in BC patients in southern Taiwan is important. |
|
Suggestion 3 |
The control group received usual care. Therefore the specific contribution of CBT for symptom reduction remains unclear. Some kind of "placebo" treatment (e.g., support group meetings without the specific elements of CBT) would have helped to control the unspecific effects of being included in a treatment program, receiving attention and the possibility to talk about one's problems etc. This shortcoming should be addressed by the authors. |
|
Response 3 |
Thanks for the comments. The control group has a usual care by nurses’ health education. For example, when patient complained about hot flushes and night sweats, nurses may give suggestions to deal with the symptoms. |
|
Suggestion 4 |
For sleep disturbances, anxiety and depression, treated patients did not differ from the controls at T1 (post-test), but only later, at 1 month and 3 months post-test. What could be the reason for this delayed response to treatment? |
|
Response 4 |
Thanks for the comments. According to a study by Peoples et al., [28], CBT for insomnia could manage sleep disturbances and thus reduce depression in cancer survivors (line377-378). In our study, sleep disturbances were decreased at T1, which later improved depression and anxiety later on. |
|
Suggestion 5 |
Was CBT administered by nurses without support by a qualified psychotherapist? Please clarify! |
|
Response 5 |
Thanks for the suggestion. The CBT was administered a licensed psychotherapist and an experienced registered nurse. The statement was elaborated on page 2. |
|
Minor comments: |
|
|
Suggestion 6 |
l. 98: depression/anxiety/sleep quality scores: which instruments were they based on? |
|
Response 6 |
Thanks for the comments. 2) had an anxiety or depression HADS score of 7 or above and a sleep quality PSQI score higher than 5. (line 100) |
|
Suggestion 7 |
l. 215: what does "AR(1)" stand for? |
|
Response 7 |
Thanks for the comments. AR(1) stands for First-order autoregressive. We selected first-order autoregressive AR(1) (line 225). |
|
Suggestion 8 |
Results: for the analyses of the main outcome measures, testing for interactions between the factors "group" and "time point" would be appropriate and informative. |
|
Response 8 |
Thanks for the comments. In this study, group referred as experimental group receiving the intervention of CBTM vs. control group receiving usual healthcare education. Four times point data were collected: baseline, T1 (post-intervention), T2 (one-month follow-up), and T3 (three months follow-up), which were collected at different interval duration (i.e., three months, one month, and three months intervals). We proposed to exam the effect of interaction effect and intervention*duration. We employed GEE to estimate T3 outcomes (PSQI, anxiety, depression, MQOL, and PQOL) using the data of baseline, T1, and T2 (see Table 3). The CBTM group for the PSQI estimated score at T3 was 3.56 (intercept) + 3.27 (CBTM baseline) + (-4.15) (CBTM group*baseline) + (-2.43) (CBTM group*T1) + (-.87) (CBTM*T2) = -.62. Control group, on the other hand, the estimated score of the PSQI at T3 was 3.56 + 3.94 (control group baseline) + 1.78 (T1) + .78 (T2) = 10.06. (line 329-333) |
|
Suggestion 9 |
A measure of variance (standard deviation) should be added to the graphs in Figures 2-5. |
|
Response 9 |
Thanks for the comments. We have tried to add the SD (standard deviation) in the graphs in figures 2-5. But we found the readability greatly decrease. Therefore, we have put the SD in the figures instead which also have same effect. |
|
Suggestion 10 |
Table 2 / Discussion (l. 373): According to the Table, there were no significant group differences at any time point for PQOL. However, the contrary is stated in the Discussion ("PQOL was significantly greater in the experimental group..."). Please clarify! |
|
Response 10 |
Thanks for the comments. In Table 2, the mean difference of PQOL was examined between experimental and control groups from baseline, T1, T2, and T3, and the results showed insignificant difference over time. However, in Table 3, the interaction effect between intervention and time was examined in GEE, and the results show significant effect at T1 and T2. We added “the GEE result of Table 3 (line 391)”. |
|
Language: |
|
|
Suggestion 11 |
l. 22: "over times" should be "over time" |
|
Response 11 |
Thanks for the suggestion. Corrected as suggested. |
|
Suggestion 12 |
l. 252 and several other places in the Results section: "as showed" should be "as shown", "a stead reduction" should be "a steady reduction" |
|
Response 12 |
Thanks for the suggestion. Corrected as suggested. |
|
Suggestion 13 |
l. 253 here "steady reduction" does not apply to the control group. |
|
Response 13 |
Thanks for the suggestion. We have deleted the description about the control group. |
|
Suggestion 14 |
l. 264, l. 267 and several other places in the Results section: "in baseline", "in T2"etc should be "at baseline", "at T2" etc. |
|
Response 14 |
Thanks for the suggestion. Corrected as suggested. |
|
Suggestion 15 |
l. 316 and several other places in the Results section: "slop" should be "slope" |
|
Response 15 |
Thanks for the suggestion. Corrected as suggested. |
|
Suggestion 16 |
l. 343 "...physical health well-being" should be "...physical well-being" |
|
Response 16 |
Thanks for the suggestion. Corrected as suggested. |
|
Suggestion 17 |
l. 371: "strongly significantly better" sounds odd |
|
Response 17 |
Thanks for the suggestion. We have revised to “significantly improved” |
|
Suggestion 18 |
l. 381: "manipulate" should probably be "control" |
|
Response 18 |
Thanks for the suggestion. Corrected as suggested. |

Reviewer 3 Report
In this work, authors present a study aimed to examine the effectiveness of CBTM for sleep disturbances and others mental health symptoms among patients with BC. Several psychopathological measures are assessed and the results are positive, nevertheless, they carried out the analysis without controlling possible confounding factors. I find this paper interesting due to provide relevant information specifically to women who suffer breast cancer. However, some suggestions are proposed to improve the work.
- In the introduction section (page 2, line 76), you write “that a combined CBTM…” but you don't name any “combined” referring CBTM along the manuscript again. Is it maybe a typo?
- In materials and methods section, you describe that the study design is “A nonequivalent, control group, pretest-posttest pilot study was conducted…” but then, in the design: “A randomized controlled experimental study design was used to investigate the efficacy of CBTM for BC patients who met the inclusion criteria”. Please, clarify this part.
- In participant’s section, one eligibility criteria were 2) had an anxiety or depression score of 7 or above and a sleep quality score higher than 5. According which instrument? Please, consider referencing it.
- What sociodemographic data were collected? Please, consider include a short paragraph in “Data collection” section explaining this information.
- Did you use the original instruments or the specific version of your language?
- In the Health Survey, the higher the score on the scale, the better health? Please, clarify it in the paragraph to a better understanding of the results.
- The CBTM protocol was very well explained, but… who conducted the sessions? A nurse or a psychologist? (page 5, line 190-192). Was a specific training done previously? where were the sessions conducted? Please, consider to include this information in this section.
- What were the reasons of the participants to lost to follow up? Please, provide this information in the figure 1 or in the first paragraph of the results.
- Footnotes are necessary in table 2 (SD, CBTM, C.I) and table 3.
- Discussion is well written and compares the results with previous studies, but I consider that this section needs to emphasize on the interpretation of the results that the authors have obtained and to reflect why they have obtained these results (specifically the results of PQOL).
- In the limitation section (page 14, line 379) you write “Physiological and self-measures…”. Which physiological data did you assess?
- The conclusions section is a bit long and repetitive. Please, consider summarized it a bit.
Author Response
Table of Responses to Comments by Reviewer 3
|
Suggestion 1 |
In the introduction section (page 2, line 76), you write “that a combined CBTM…” but you don't name any “combined” referring CBTM along the manuscript again. Is it maybe a typo? |
|
Response 1 |
The CBTM stands for cognitive behavioral therapy plus coping management. The original meaning of “combined CBTM” is to emphasize that CBTM is the combination of CBT and coping management. Since this might cause misunderstanding, we delete the word “combined”. Thanks for your suggestion. |
|
Suggestion 2 |
In materials and methods section, you describe that the study design is “A nonequivalent, control group, pretest-posttest pilot study was conducted…” but then, in the design: “A randomized controlled experimental study design was used to investigate the efficacy of CBTM for BC patients who met the inclusion criteria”. Please, clarify this part. |
|
Response 2 |
Thanks for your suggestion. In order not to cause confusion, we have combined “Study design” and “Design” into one section. |
|
Suggestion 3 |
In participant’s section, one eligibility criteria were 2) had an anxiety or depression score of 7 or above and a sleep quality score higher than 5. According which instrument? Please, consider referencing it. |
|
Response 3 |
Thanks for suggestion. We have added the name of measurement tools to the description. 2) had an anxiety or depression HADS score of 7 or above and a sleep quality PSQI score higher than 5. |
|
Suggestion 4 |
What sociodemographic data were collected? Please, consider include a short paragraph in “Data collection” section explaining this information. |
|
Response 4 |
Thanks for the suggestion. We have added a short paragraph to explain the sociodemographic data collected on page 3 as follows: “The sociodemographic information questionnaire was used to measure individual’s characteristics. Data was collected on age; education reported as illiterate, elementary, junior high, senior high and college; income reported in New Taiwan currency (NT dollars); adjuvant therapy; breast cancer stage classified into four stages; disease characteristics measured as duration in months.” |
|
Suggestion 5 |
Did you use the original instruments or the specific version of your language? |
|
Response 5 |
We use the original instruments with proper Chinese translation. |
|
Suggestion 6 |
In the Health Survey, the higher the score on the scale, the better health? Please, clarify it in the paragraph to a better understanding of the results. |
|
Response 6 |
Thanks for the suggestion. We added the sentence “The higher the score on the scale, the better health” (line 138-139). |
|
Suggestion 7 |
The CBTM protocol was very well explained, but… who conducted the sessions? A nurse or a psychologist? (page 5, line 190-192). Was a specific training done previously? where were the sessions conducted? Please, consider to include this information in this section. |
|
Response 7 |
Thanks for the positive comments. These twelve sessions were conducted by a licensed psychotherapist and an experienced registered nurse. The description was added in the manuscript on page 4. |
|
Suggestion 8 |
What were the reasons of the participants to lost to follow up? Please, provide this information in the figure 1 or in the first paragraph of the results. |
|
Response 8 |
Thanks for the comments. The reason for participants were lost in the follow-up were lost of contact or unwilling to complete the questionnaires. |
|
Suggestion 9 |
Footnotes are necessary in table 2 (SD, CBTM, C.I) and table 3. |
|
Response 9 |
Thanks for the suggestion. We have added the footnote to table 2 and 3. |
|
Suggestion 10 |
Discussion is well written and compares the results with previous studies, but I consider that this section needs to emphasize on the interpretation of the results that the authors have obtained and to reflect why they have obtained these results (specifically the results of PQOL). |
|
Response 10 |
Thanks for the comments. For the PQOL outcome in Table3, the control group revealed a significant higher score (B =-3.53, p < .001) compared with CBTM group (B =-3.95, p = .19) at baseline. The slop of PQOL mean score gradually declined from baseline to T1 (B =-2.53, p = .002) and T2 (B =-.58, p = .47). Compared with CBTM group, the intervention effect was B =6.97 at T1 (CBTM group*baseline, p < .001) and B =2.99 at T2 (CBTM*T2, p = .009). This corresponds with data in Figure 5 that PQOL declines over time in control group, while improves continuously in CBTM group. Simply put, the estimated PSQI at T3 for CBTM group was 59.28 + (-3.95) (CBTM baseline) + 6.97 (CBTM group*baseline) + 2.99 (CBTM group*T1) + 1.64 (CBTM group*T2) = 66.93. The estimated PSQI at T3 for control group was 59.28 + (-3.53) (control baseline) + (-2.53) (T1) + (-.58) (T2) = 52.64. (line 395-404) |
|
Suggestion 11 |
In the limitation section (page 14, line 379) you write “Physiological and self-measures…”. Which physiological data did you assess? |
|
Response 11 |
Thanks for the comments. Physiological data that we assess are: “Adjuvant therapy” & “stage of cancer” |
|
Suggestion 12 |
The conclusions section is a bit long and repetitive. Please, consider summarized it a bit. |
|
Response 12 |
We have rewritten the content and make it concise. |
